

# Effects of preservation duration at 4 °C on the quality of RNA in rabbit blood specimens

Jiaojiao Song[1,2] and Junmei Zhou[1,2]

[1] Children's Hospital of Shanghai, Shanghai, China
[2] Children's Hospital Affiliated to Shanghai Jiao Tong University, Shanghai, China

## ABSTRACT

A prolonged preservation duration of blood specimens at 4 °C may occur due to the distance from collection points to storage facilities in many biobanks, especially for multicenter studies. This could lead to RNA degradation, affecting downstream analyses. However, effects of preservation durations at 4 °C on RNA quality in blood specimens need to be studied. We collected rabbit blood using EDTA tubes and stored them at 4 °C for different preservation durations. Then, we examined the quality of RNA from whole blood and leukocytes isolated from rabbit blood. Our results show that the purity of whole blood RNA and leukocyte RNA does not indicate significant change after rabbit blood is stored at 4 °C for different preservation durations (from 1 h to 7 days). The integrity of leukocyte RNA indicates the same result as above, but the integrity of whole blood RNA is significantly decreased after rabbit blood is stored at 4 °C for over 3 days. Moreover, expression of *SMAD7*, *MKI67*, *FOS*, *TGFβ1* and *HIF1α* of whole blood RNA and leukocyte RNA remains basically stable, but *PCNA* expression of whole blood RNA or leukocyte RNA is significantly decreased after rabbit blood is stored at 4 °C for over 24 h or 7 days. Therefore, these results suggest that high-quality RNA is obtained from the fresher blood specimens and if blood specimens are stored for over 3 days at 4 °C, the quality of leukocyte RNA is more stable and of better quality than that of whole blood RNA.

## INTRODUCTION

As a valuable resource for high quality biological specimens and associated data, the biobank is considered to be a bridge linking basic medical research and translational research (*Baker, 2012*). In many biobanks, one of the routinely collected biospecimens is human blood, which is stored for scientific research, medical applications and diagnostic purposes. With the development of '-omics' technologies, blood specimens are increasingly used as a less invasive and valuable source of RNA for profiling gene expression (*Barnes et al., 2010*; *Hebels et al., 2013*). However, a major limitation of blood RNA studies in the biobank is RNA degradation by abundant and ubiquitous RNases that are present during blood collection and preservation. Thus, understanding the effects

Corresponding author
Jiaojiao Song, jjiaosong@163.com

of pre-analytical variables of blood specimen preservation on RNA quality is one of the critical aspects of biobanking.

In many biobanks, human blood specimens are collected by clinical departments and then transported and stored in the biobank (*Huang et al., 2017*; *Malentacchi et al., 2016*), so a prolonged preservation duration at 4 °C may occur due to the distance from collection points to storage facilities, especially for multicenter studies. A longer preservation duration before freezing in the biobank is believed to have significant effects on the quality of blood specimens and could consequently lead to RNA degradation, affecting downstream analyses (*Wang et al., 2018*). To ensure the quality and stabilization of blood specimens and minimize the degradation of blood RNA, PaxGene Blood RNA tubes and Tempus tubes containing RNA stabilizing reagents are recommended as alternatives for blood specimens collection (*Chai et al., 2005*; *Duale et al., 2012*). However, these RNA collection tubes are quite expensive and require dedicated kits for RNA extraction. Therefore, most of the external quality assessment protocols adapt ethylenediaminetetraacetic acid (EDTA) blood collection tubes for their gene expression studies (*Malentacchi et al., 2014*; *Pazzagli et al., 2013*). In addition, the EDTA tube is one of the most widely and commonly used blood specimen collection tubes in clinical sample collection (*Nilsson et al., 2005*), and there may be a need to perform RNA expression profiling experiments with legacy blood specimens that were initially collected by EDTA tubes (*Thorn et al., 2005*). These EDTA additives inhibit clotting but do little to maintain the gene expression profiles or control other pre-analytical variables of specimen processing (i.e., a prolonged preservation duration at 4 °C). Therefore, it is important to examine the effects of preservation durations at 4 °C on the quality of RNA from blood specimens that are collected using EDTA tubes, but there is a lack of research in this regard.

In addition, blood specimens may be processed to produce aliquots of leukocytes, red blood cells (RBC), serum and blood platelets. Among these, the leukocytes are the common research targets. A previous report has suggested that in RNA expression profiling experiments, there is increased noise and reduced signal derived from whole blood compared with a leukocyte isolation protocol (*Feezor et al., 2004*). So, it is important to isolate leukocyte RNA from the blood specimens during the preservation procedure. Therefore, in this current study, we observed the effects of preservation duration at 4 °C on the quality of RNA, not only from the whole blood but also from the leukocytes.

With regard to the assessment of RNA quality, we used the method confirmed by the International Society for Biological and Environmental Repositories RNA Proficiency Testing Program. In the current study, we collected rabbit blood specimens with EDTA tubes and stored split aliquots at 4 °C for different preservation durations. As for the reason we used rabbit blood specimens instead of human blood specimens in this study, it's that previous reports had shown that our biobank was established to focus on pediatric diseases (*Gao et al., 2015*), and human blood specimens collected from children are very valuable in a biobank related to pediatric diseases (*Giesbertz, Bredenoord & Van Delden, 2015*). Therefore, to observe the effects of preservation durations at 4 °C of blood specimens on the quality of RNA from the whole blood and the leukocytes, we examined

the purity, the integrity and gene expression of whole blood RNA and leukocyte RNA, after rabbit blood specimens were stored at 4 °C for different preservation durations.

## MATERIALS AND METHODS

### Experimental animals

Healthy adult New Zealand white rabbits were housed individually in a 12 hours (h) light/dark cycle in a temperature- and humidity-controlled environment, with rabbit chow and tap water freely available. All experimental procedures were performed in compliance with the guidelines on the ethical use of animals in Shanghai Jiao Tong University School of medicine DLAS (2019-B-033), as well as in conformance to international guidelines. Moreover, the acceptable euthanasia methods for the rabbits are carbon dioxide asphyxiation and pentobarbital overdose. In order to minimize pain and distress, animals meeting any one of the following criteria were euthanized prior to the planned end of the experiments: ≥20% body weight loss, inability or extreme reluctance to stand persisting for 24 h, rectal body temperature <91.4 °C, severe acute anemia (<15% hematocrit or red blood cell (RBC) $<3 \times 10^6$ cells/μL), or other signs of severe organ system dysfunction (*Jackson et al., 2019*). All efforts were made to minimize animal suffering and reduce the number of animals used.

### Blood collection and processing

Blood specimens were collected from the marginal ear vein of 12 healthy rabbits, by using EDTA tubes. The needle attached to EDTA tube was inserted to marginal ear vein about two-thirds of the distance from the head to the tip of the ear. Once rabbit blood specimens were collected, each tube was gently inverted 6–8 times. After the collection of rabbit blood specimens, the rabbits were returned to their home cages and continued to be fed with rabbit chow and tap water freely available, so above euthanasia methods were not performed for the rabbits in this study. Moreover, the rabbits in this study were not used for successive blood sampling.

The rabbit blood specimens were aliquoted into one ml per tube and stored at 4 °C for 1 h, 2 h, 8 h, 16 h, 24 h, 3 days (d), 5 days and 7 days until being used in the procedures that followed. For the analysis of whole blood RNA, rabbit blood specimens stored at 4 °C for different preservation durations (from 1 h to 7 days) were used to extract RNA. For the analysis of leukocyte RNA, rabbit blood specimens stored at 4 °C for different preservation durations (from 1 h to 7 days) were used to isolate leukocytes, and then the isolated leukocytes were used to extract the RNA.

The leukocytes were isolated from blood specimens by using the RBC lysis method (*Heng, Ruan & Gan, 2018*). Briefly, different blood aliquots were centrifuged at 1,600*g* for 10 minutes (min) at 4 °C and removed the supernatant. The precipitate in each tube was added to 3 volumes of RBC lysis buffer. Each tube was gently inverted and incubated at room temperature for 5 min. When the solution in each tube was clear and transparent, it was centrifuged at 3,000*g* for 10 min at 4 °C and the supernatant was removed. After those steps, the precipitate in each tube was added to 2 volumes of RBC lysis buffer and

centrifuged at 3,000$g$ for 10 min at 4 °C. After removing the supernatant, one ml PBS was added to dissolve the precipitate which contained the leukocytes.

## RNA extraction and quality analysis

Whole blood RNA and leukocyte RNA were extracted using the TRIzol reagent single-step method as described previously (*Chomczynski & Sacchi, 2006*). Each whole blood or leukocyte aliquot was added to 3 volumes of TRIzol reagent (Invitrogen, Stockholm, Sweden). The specimens were incubated in TRIzol reagent at room temperature for 5 min. Afterwards, to each ml of TRIzol reagent in each tube was added 200 µl chloroform and the tubes were shaken for 15 seconds (s) before being kept at room temperature for 3 min. The tubes were centrifuged at 13,400$g$ for 15 min at 4 °C, after which the upper colorless aqueous phase was transferred to a new RNAse-free tube. Then, to each tube was added a same volume of isopropyl alcohol and incubated at room temperature for 10 min. The tubes were centrifuged at 13,400$g$ for 10 min at 4 °C, and the supernatant was removed. For washing the RNA pellet, one ml 75% ethanol was added to each tube. The tubes were centrifuged at 8,375$g$ for 10 min at 4 °C and the supernatant was removed. Finally, the RNA pellet was air-dried and dissolved in 20 µl RNAse-free water with diethyl pyrocarbonate (DEPC).

For the analysis of RNA quality, a NanoDrop$^{TM}$ 2000c spectrophotometer (Thermo Scientific, Waltham, MA, USA) was used to determine the RNA concentration (ng/µl) and the RNA purity, which was measured by the optical density (OD) 260/280 ratio. An Agilent 2100 bioanalyzer (Agilent Technologies, Santa Clara, CA, USA), in conjunction with the RNA 6000 Nano LabChip kit, was used to evaluate the RNA integrity, which was expressed as the RNA integrity number (RIN). The assays were performed according to the manufacturer's instructions.

## Complementary DNA synthesis and real-time quantitative PCR

Complementary DNA (cDNA) was reverse-transcribed from 1 µg RNA from the whole blood and the leukocytes using a Transcriptor First Strand cDNA Synthesis Kit (Roche, Basel, Switzerland). To yield a total volume of 20 µl, anchored-oligo (dT)$_{18}$ primer (2.5 µM), transcriptor reverse transcriptase reaction buffer (1× 8 mM MgCl$_2$), protector RNase inhibitor (20 U), deoxynucleotide mix (1 mM), transcriptor reverse transcriptase (10 U) and water (PCR-grade) were added according to the manufacturer's instructions. The reaction mixture was incubated at 55 °C for 30 min to perform the reverse transcription and then was incubated at 85 °C for 5 min. After those steps, the tubes were immediately put on ice.

To assess the gene expression in all RNA groups, the RT-qPCR reactions were performed by using the FastStart Essential DNA Green Master kit (Roche, Basel, Switzerland). The primers were designed for six genes: *SMAD7*, *MKI67*, *Fos*, *TGFβ1*, *HIF1α* and *PCNA* (BioTNT, Shanghai, China). To yield a total volume of 10 µl, one µl cDNA, one µl forward primer (10 µM), one µl reverse primer (10 µM), five µl SYBR Supermix (2×) and two µl nuclease-free water were added according to the manufacturer's instructions. The reaction mixture was initiated at 95 °C for 10 min, followed by 45 cycles

**Table 1 Primer sequences use for RT-qPCR.**

| Primer | Forward | Reverse | Length |
|--------|---------|---------|--------|
| GAPDH | TGATCCATTCATTGACCTCCAC | CGTTCTCAGCCTTGACCGT | 92 |
| SMAD7 | AGGCAGAAGAAACGGAGAG | TTACAAAGTGGAACAAGACGC | 110 |
| MKI67 | CAAGATTCCAAAGCCCATTC | GGCAACGCTGTCTTCTGAG | 120 |
| Fos | ACAGCCTCTCCTACTACCAC | AGATCCGAACAGAAGTCCT | 85 |
| TGFβ1 | GGGTCTCTTCTGTCGCTTGT | ATGTGCTCGGTGTTTACGG | 61 |
| HIF1α | CGGAGCAAAGAGTCTGAAG | ACACGCAGGTAACTGATGGT | 119 |
| PCNA | GACATCATTACACTGAGGGC | CCAAGCTGTTCAACATCTAAG | 125 |

of 95 °C for 10 s, 60 °C for 10 s and 72 °C for 10 s. All reactions were performed on the Light Cycler 96 PCR system (Roche, Basel, Switzerland). The same sample was run three different times to remove any outliers. Expression levels of each gene were calculated from the mean cycle threshold (Ct) values and were normalized to the expression of glyceraldehydes-3-phosphate dehydrogenase (GAPDH). The primer sequences for each gene are shown in Table 1.

## Statistical analysis

All data was analyzed by using Graphpad Prism 6. The data of OD 260/280 ratios, RIN values and gene expression results among the different groups were determined by one-way analysis of variance (ANOVA), which was followed by a Tukey's post hoc pairwise comparison test. For all results, $P < 0.05$ was considered statistically significant. Numerical data was expressed as the mean ± standard error of the mean (s.e.m.).

## RESULTS

### The effect of preservation durations of rabbit blood specimens at 4 °C on the purity of whole blood RNA and leukocyte RNA

To examine the effect of preservation durations of rabbit blood specimens at 4 °C on the purity of whole blood RNA and leukocyte RNA, we collected blood specimens from the rabbits using EDTA collection tubes and stored them at 4 °C for 1 h, 2 h, 8 h, 16 h, 24 h, 3 days, 5 days and 7 days, respectively. Among these, rabbit blood specimens stored at 4 °C for 1 h were used as the control, because a prolonged contemporary preservation duration at 4 °C may occur due to the distance from collection points to storage facilities in many biobanks, especially for multicenter studies. We extracted RNA from the whole blood and the leukocytes that were isolated from rabbit blood specimens and then we analyzed the purity of RNA, which was evaluated by the OD 260/280 ratio of RNA. The OD 260/280 ratio of RNA with high purity is 1.8–2.1 (Kong et al., 2018). Our results showed that the OD 260/280 ratio of whole blood RNA from rabbit blood specimens stored at 4 °C for different preservation durations (from 1 h to 7 days), ranged between 1.9 and 2.1 (Fig. 1A). The OD 260/280 ratio of leukocyte RNA from rabbit blood specimens stored at 4 °C for different preservation durations (from 1 h to 7 days), ranged between 1.8 and 2.1 (Fig. 1B). These results suggest that different preservation durations

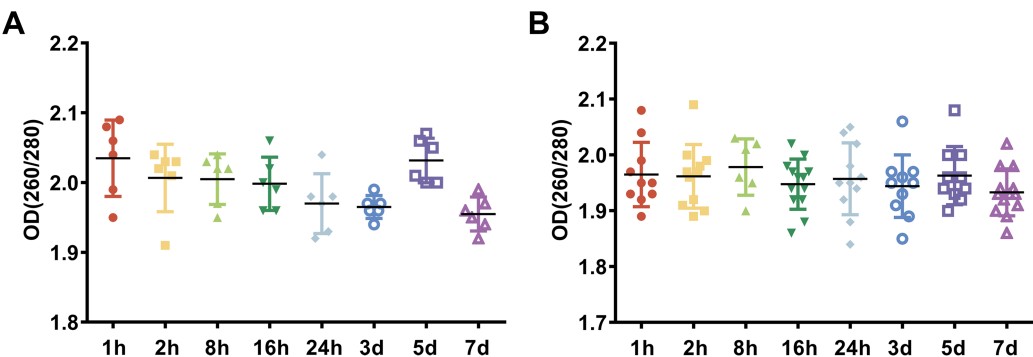

**Figure 1 The assessment of RNA purity under different preservation durations (from 1 h to 7 days) at 4 °C.** (A) The OD 260/280 ratios of whole blood RNA under different preservation durations (from 1 h to 7 days) at 4 °C. (B) The OD 260/280 ratios of leukocyte RNA under different preservation durations (from 1 h to 7 days) at 4 °C.               

(from 1 h to 7 days) of rabbit blood specimens at 4 °C have no influence on the purity of whole blood RNA or leukocyte RNA.

## The effect of preservation duration of rabbit blood specimens at 4 °C on the integrity of whole blood RNA and leukocyte RNA

To examine the effect of preservation duration of rabbit blood specimens at 4 °C on the integrity of whole blood RNA and leukocyte RNA, we analyzed the RIN value among different preservation durations (from 1 h to 7 days). A scoring system between 1 and 10 were used to evaluate the RIN values of RNA, with <5 indicating degraded RNA and >8 indicating high-quality RNA. The cutoff RIN values between 6 and 7 were used in gene expression array analyses (*Wang et al., 2015*). In this study, a RIN value of 6.5 was chosen as a cutoff for RNA quality (*Kap et al., 2014*). After rabbit blood specimens were stored at 4 °C for different preservation durations (from 1 h to 7 days), the RIN value of whole blood RNA specimens showed a significant difference (Figs. 2A–2H). After rabbit blood specimens were stored at 4 °C for 1 h, 2 h and 8 h, the mean RIN values of whole blood RNA were 8.55 ± 0.28, 8.22 ± 0.53 and 8.40 ± 0.14, respectively, which were all above 8. After rabbit blood specimens were stored at 4 °C for 16 h and 24 h, the mean RIN values of whole blood RNA were 6.96 ± 0.92 and 7.05 ± 1.25, which were below 8, but above the cutoff value 6.5. After rabbit blood specimens were stored at 4 °C for 3 days, 5 days and 7 days, the mean RIN values of whole blood RNA were respectively 5.43 ± 0.73, 5.45 ± 0.57 and 5.20 ± 0.36, which were below the cutoff value 6.5 ($P < 0.01$, Fig. 2I). However, after rabbit blood specimens were stored at 4 °C for different preservation durations (from 1 h to 7 days), the RIN values of leukocyte RNA showed no change (Figs. 3A–3H). The mean RIN values of leukocyte RNA after rabbit blood specimens were stored at 4 °C for different preservation durations (from 1 h to 5 days) were above 8 (for 1 h: 8.69 ± 0.22, for 2 h: 8.86 ± 0.18, for 8 h: 9.06 ± 0.15, for 16 h: 8.43 ± 0.25, for 24 h: 8.38 ± 0.50, for 3 days: 8.62 ± 0.15, for 5 days: 8.41 ± 0.25, $P = 0.11$, Fig. 3I), while that for 7 days was 7.88 ± 0.30, which was below 8, but above the cutoff value 6.5. These results suggest that different preservation durations (from 1 h to 7 days) of rabbit blood

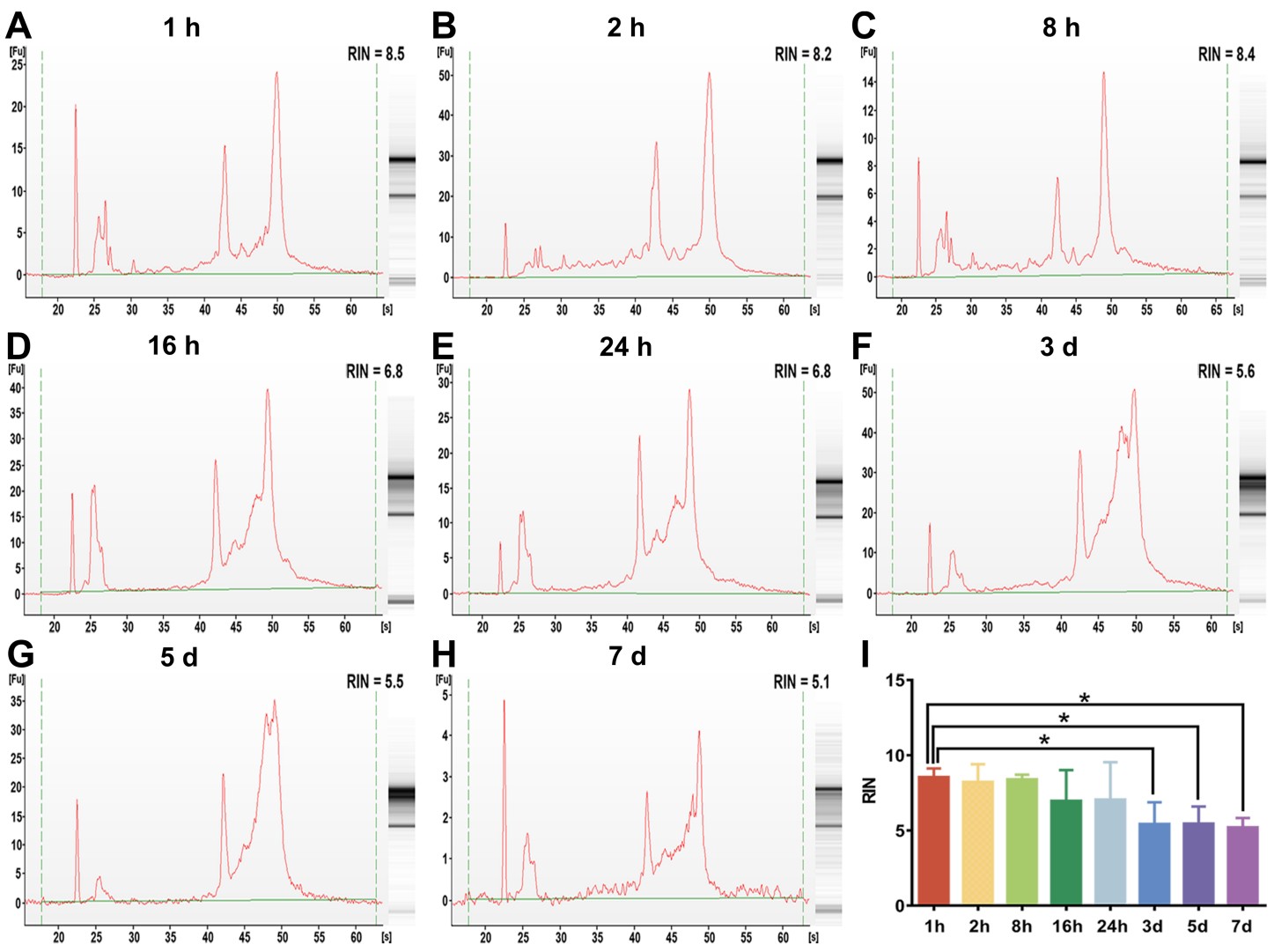

**Figure 2 The assessment of whole blood RNA integrity under different preservation durations (from 1 h to 7 days) at 4 °C.** (A–H) The representative RIN values of whole blood RNA under different preservation durations (from 1 h to 7 days) at 4 °C of 1 h, 2 h, 8 h, 16 h, 24 h, 3 days, 5 days and 7 days, respectively. (I) The mean RIN values of whole blood RNA under different preservation durations (from 1 h to 7 days) at 4 °C ($n = 6$, one-way ANOVA, $^*P < 0.01$). Data are shown as the mean ± s.e.m.

specimens at 4 °C have no influence on the integrity of leukocyte RNA, but the integrity of whole blood RNA is decreased after such rabbit blood specimens are stored for more than 3 days at 4 °C.

## The effect of preservation durations of rabbit blood specimens at 4 °C on gene expression levels of whole blood RNA and leukocyte RNA

To examine whether the preservation durations of rabbit blood specimens at 4 °C induced changes in gene expression, we analyzed specific gene expression of whole blood RNA and leukocyte RNA by RT-qPCR. The specific genes described in the literature were selected to represent different cellular regulatory pathways, such as *SMAD7*, a marker of signal transduction molecule; *MKI67*, a marker of proliferation *Ki-67*; *Fos*, a marker of

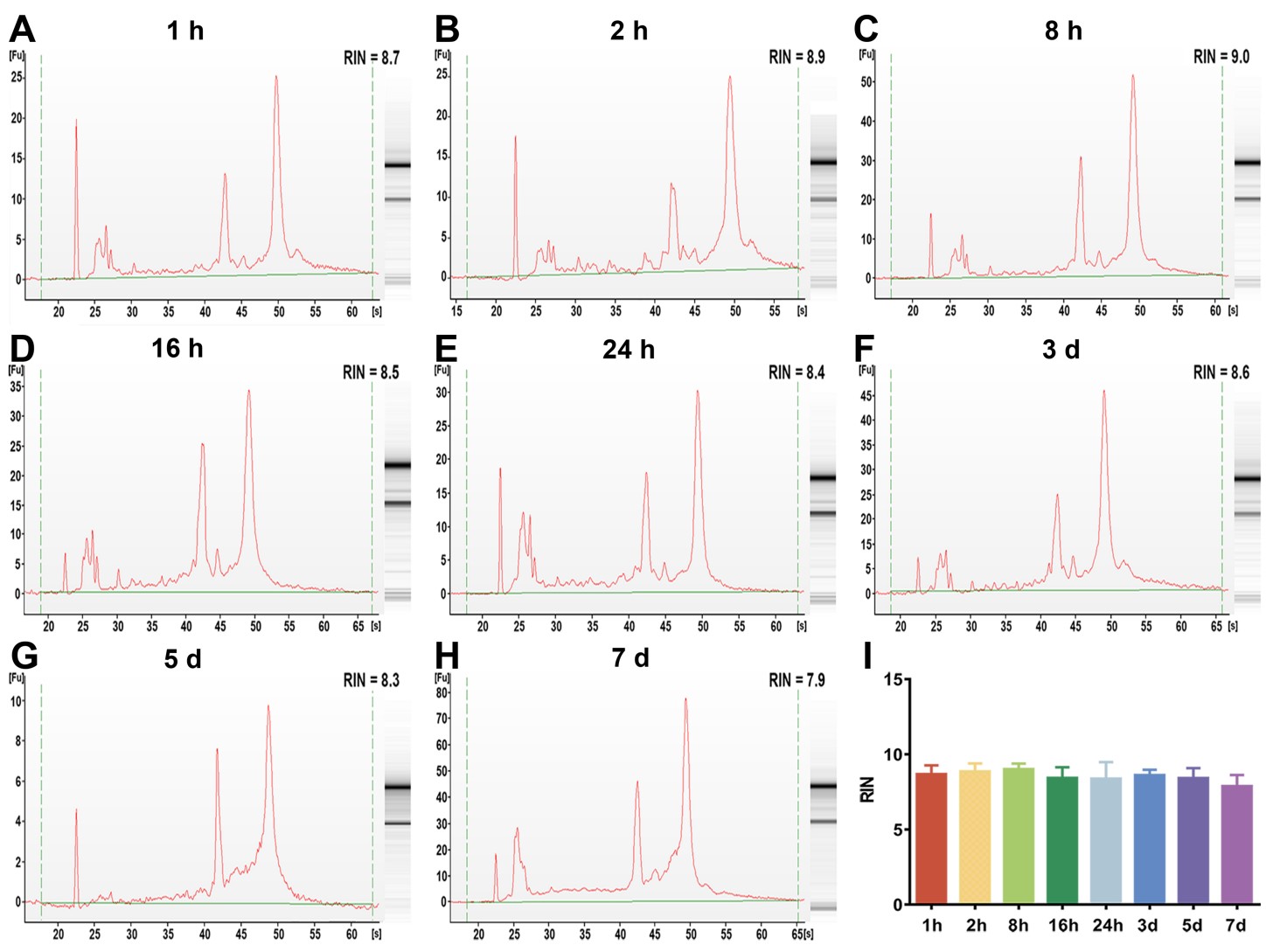

**Figure 3 The assessment of leukocyte RNA integrity under different preservation durations (from 1 h to 7 days) at 4 °C.** (A–H) The representative RIN values of leukocyte RNA under different preservation durations (from 1 h to 7 days) at 4 °C of 1 h, 2 h, 8 h, 16 h, 24 h, 3 days, 5 days and 7 days, respectively. (I) The mean RIN values of leukocyte RNA under different preservation durations (from 1 h to 7 days) at 4 °C ($n = 7$, one-way ANOVA, $P = 0.11$). Data are shown as the mean ± s.e.m.               

immediate early response; *TGFβ1*, a marker of growth factor; *HIF1α*, a marker of hypoxia response; and *PCNA*, a marker of cell proliferation (*Micke et al., 2006*; *Wang et al., 2015*). Figure 4 shows the mean Ct value of in six specific genes classified by different preservation durations (from 1 h to 7 days). We see that after rabbit blood specimens were stored at 4 °C for different preservation durations (from 1 h to 7 days), the analyzed marker genes of *SMAD7*, *MKI67* and *Fos* remained basically stable at the level of whole blood RNA and leukocyte RNA (Figs. 4A–4F). The analyzed marker genes of *TGFβ1* and *HIF1α* also remained basically stable at the level of whole blood RNA, but they were significantly increased at the level of leukocyte RNA after rabbit blood specimens were stored at 4 °C for 2 h (Figs. 4G–4J). Moreover, the most significant changes in gene expression patterns were observed in the analyzed marker gene *PCNA*, which had a
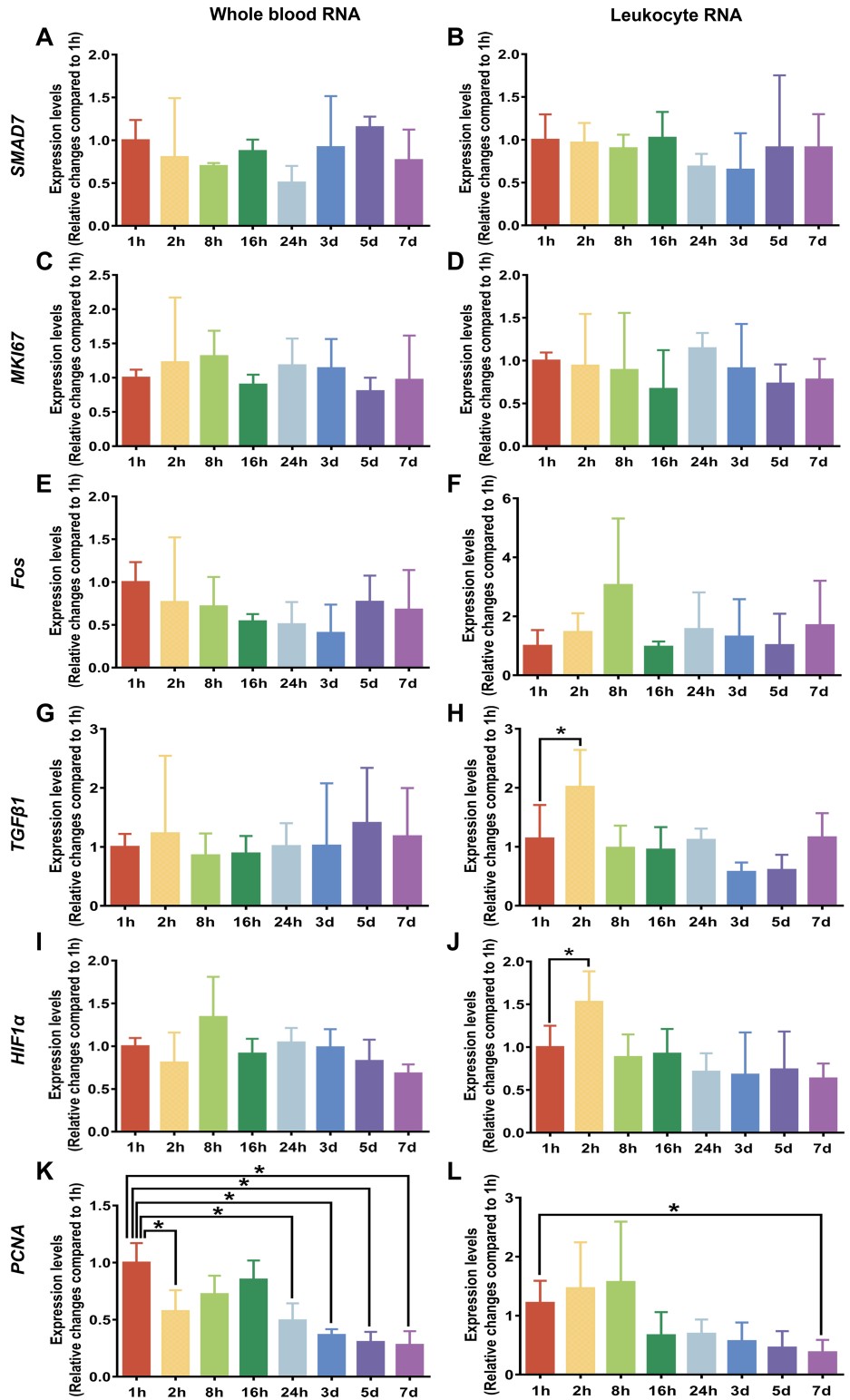

**Figure 4 Gene expression levels under different preservation durations (from 1 h to 7 days) at 4 °C.**
(A) The mean Ct values of *SMAD7* gene of whole blood RNA ($n = 3$, one-way ANOVA, $P = 0.57$).
(B) The mean Ct values of *SMAD7* gene of leukocyte RNA ($n = 4$, one-way ANOVA, $P = 0.85$). (C) The mean Ct values of *MKI67* gene of whole blood RNA ($n = 3$, one-way ANOVA, $P = 0.09$). (D) The mean

**Figure 4** (continued)
Ct values of *MKI67* gene of leukocyte RNA ($n = 3$, one-way ANOVA, $P = 0.89$). (E) The mean Ct values of *Fos* gene of whole blood RNA ($n = 4$, one-way ANOVA, $P = 0.52$). (F) The mean Ct values of *Fos* gene of leukocyte RNA ($n = 5$, one-way ANOVA, $P = 0.20$). (G) The mean Ct values of *TGFβ1* gene of whole blood RNA ($n = 3$, one-way ANOVA, $P = 0.99$). (H) The mean Ct values of *TGFβ1* gene of leukocyte RNA ($n = 5$, one-way ANOVA, $^{*}P < 0.001$). (I) The mean Ct values of *HIF1α* gene of whole blood RNA ($n = 3$, one-way ANOVA, $P = 0.16$). (J) The mean Ct values of *HIF1α* gene of leukocyte RNA ($n = 6$, one-way ANOVA, $^{*}P < 0.001$). (K) The mean Ct values of *PCNA* gene of whole blood RNA ($n = 3$, one-way ANOVA, $^{*}P < 0.001$). (L) The mean Ct values of *PCNA* gene of leukocyte RNA ($n = 6$, one-way ANOVA, $^{*}P < 0.001$). Data are shown as the mean ± s.e.m.

dramatic downregulation of whole blood RNA after rabbit blood specimens were stored at 4 °C for more than 24 h and a dramatic downregulation of leukocyte RNA after rabbit blood specimens were stored at 4 °C for 7 days (Figs. 4K and 4L).

## DISCUSSION

Blood specimens, as the fundamental part of a biobank, offer a readily accessible valuable RNA source for further use in research and diagnostic procedures. Among all fractions of whole blood, the leukocytes in most cases, can be used for RNA expression profiling studies (*Heng, Ruan & Gan, 2018*). Although the fresher the sample the better is the golden rule when working with blood specimens, a prolonged preservation duration at 4 °C may occur due to the distance from collection points to storage facilities in many biobanks, which could lead to RNA degradation and affecting downstream analyses. Therefore, evaluating the effects of preservation durations at 4 °C for blood specimens is critical to ensure high-quality RNA from whole blood and leukocyte samples.

In this study, we collected blood specimens from rabbits using EDTA tubes and stored them at 4 °C for different preservation durations. To examine the effects of preservation durations at 4 °C of blood specimens on the quality of RNA from the whole blood and the leukocytes, we simultaneously examined the purity, the integrity and gene expression of whole blood RNA and leukocyte RNA. The main findings of our study suggest that high-quality RNA is obtained from the fresher blood specimens, and if blood specimens are stored for more than 3 days at 4 °C, the quality of leukocyte RNA is more stable and of better quality than that of whole blood RNA.

As for the preservation duration design of collected blood specimens, it is based on the current situation of the distance of the blood specimen collection points to the storage facilities. When blood specimens stored for different preservation durations (from 1 h to 7 days) at 4 °C were used to isolate the leukocytes, the analytical question was whether the preservation duration of blood specimens had any effect on the quality of the leukocytes. Based on the evidence that storing blood at 4 °C for no more than 24 h does not have an effect on the quantity and viability of leukocytes (*Zhou et al., 2015*), and our results that different preservation durations (from 1 h to 7 days) of rabbit blood specimens at 4 °C have no influence on the purity or integrity of leukocyte RNA, we propose that the preservation durations (from 1 h to 7 days) of blood specimens have no influence on the purity and the integrity of the leukocytes in this study.

RNA is considered a most fragile molecule, and RNA quality has been regarded as the criterion to measure the quality of blood specimens. Once RNA is extracted from a blood biospecimen, it's necessary to measure OD at multiple wavelengths as a basic assessment of quantity and quality of RNA specimens (*Fleige & Pfaffl, 2006*). The OD 260/280 ratio >1.8 is generally accepted as pure RNA, which is suitable for gene expression analysis; and the OD 260/280 ratio <1.8 generally indicates the presence of contaminants (*Becker et al., 2010*). In our current study, the OD 260/280 ratios of whole blood RNA and leukocyte RNA for different preservation durations (from 1 h to 7 days) at 4 °C are greater than 1.8. Therefore, our results suggest that different preservation durations (from 1 h to 7 days) at 4 °C have no influence on the purity of whole blood RNA or leukocyte RNA. Although OD values are useful to determine the approximate quantity and purity of RNA, they do not indicate the physical integrity (lack of fragmentation) (*Shabihkhani et al., 2014*). The RIN value, which contains 10 features of RNA electropherograms, has proven to be a better indicator of RNA quality than the OD ratio (*Imbeaud et al., 2005*; *Schroeder et al., 2006*). It has also been shown that specimens with a RIN value >6.5 work well in highly demanding gene array assays, while those with a RIN value between 5.0 and 6.5 could be used for RT-qPCR research, and those with a RIN value between 1 and 4.9 may not work for most studies. In this study, a RIN value of 6.5 was chosen as the cutoff value. This part of our study (see Figs. 2 and 3) showed that if blood specimens are stored for more than 3 days at 4 °C, the integrity of leukocyte RNA is better than that of whole blood RNA, and that leukocyte RNA rather than whole blood RNA can be used for gene array assays.

However, even if the purity and the integrity of RNA showed no change, it was possible that gene expression may change after blood specimens were stored for different preservation durations (from 1 h to 7 days) at 4 °C. This could result from the fact that RNA integrity, evaluated by the RIN value using the Agilent Bioanalyzer, is mainly based on rRNA features and does not represent a specific quality analysis of mRNA, that is a target for gene expression studies (*Malentacchi et al., 2016*). In order to observe the variation of whole blood RNA and leukocyte RNA in response to different preservation durations in this study, six target genes were selected to represent different cellular regulatory pathways as described in the literature (*Micke et al., 2006*; *Wang et al., 2015*). These selected genes are known to be induced or repressed by ex vivo blood handling and can be the biomarkers of blood mRNA for monitoring changes in blood specimens after collection and preservation (*Jin et al., 2013*; *Li et al., 2019*; *Malentacchi et al., 2016*). Previous research has shown that pre-analytical factors can affect very sensitive analytical procedures of RNA, such as RT-qPCR (*Nussbaumer, Gharehbaghi-Schnell & Korschineck, 2006*). In this study, the gene expression of *TGFβ1* and *HIF1α* revealed relevant changes of expression at the level of leukocyte RNA after a preservation duration for 2 h at 4 °C. These unexplained changes in gene expression seem to be associated with suboptimal blood volumes collected in the tubes (*Duale et al., 2012*). This is supported by a previous study, which suggested the fluctuation of *TGFβ1* is due to handling of blood specimens (*Zhao et al., 2012*). As for the results with other genes, they are consistent with other studies showing that the expression levels remained stable during storage on ice

over a longer period of time (*Micke et al., 2006*). In addition, our results showed that the expression of *PCNA* had a significant down-regulation at the level of whole blood RNA or leukocyte RNA, after rabbit blood specimens were stored for more than 24 h or 7 days, respectively. PCNA, an endogenous nuclear protein essential to identify replicating cells, is synthesized in early G1 and S phases of the cell cycle, and is a useful tool for the study for all these in cell cycle progressions (*Essers et al., 2005*; *Franco et al., 2010*). A previous report had suggested that gene expression of *PCNA* at different preservation durations in tonsil tissue essentially was stable when biospecimens were kept on ice (*Micke et al., 2006*), but our results with blood RNA were not in agreement. As for the reason for the down-regulation of *PCNA* expression in this study, it appears more likely that some mRNA degradation can take place during the preservation duration, so we propose that gene expression of *PCNA* may be influenced by the preservation duration of blood specimens at 4 °C. Therefore, the fresher blood specimens should be used when *PCNA* expression needs to be examined in the future specific studies.

## CONCLUSIONS

After rabbit blood specimens are collected and preserved for different preservation durations (from 1 h to 7 days) at 4 °C, the purity of whole blood RNA and leukocyte RNA are comparable, but the integrity of whole blood RNA is degraded after a preservation duration for more than 3 days at 4 °C. Moreover, *PCNA* expression of whole blood RNA and leukocyte RNA can decrease over the increased preservation duration, especially that of whole blood RNA, but the expression of other studied genes was stable. Overall, we demonstrate that high-quality RNA is obtained from the fresher blood specimens and if blood specimens are stored for more than 3 days at 4 °C, the quality of leukocyte RNA is more stable and of better quality than that of whole blood RNA. In this regard, we recommend blood specimens should be transported to biobank as soon as possible and if blood specimens are stored for more than 3 days at 4 °C, blood RNA should be extracted from the leukocytes rather than the whole blood. Moreover, expression of selected genes in this study can represent potential changes of some specific gene expression analyses, but we also recommend blood specimen collection and preservation procedures need to be studied according to specific future uses of the samples in research studies. Therefore, our results provide more information to biobank managers and also to the researchers.

### Funding

This work was supported by Shanghai Municipal Commission of Health and Family Planning (20184Y0220); Research Project of Collaborative Innovation Center of Translational Medicine (TM201827). The funders had no role in study design, data collection and analysis, decision to publish, or preparation of the manuscript.

## Grant Disclosures

The following grant information was disclosed by the authors:

Shanghai Municipal Commission of Health and Family Planning: 20184Y0220.
Research Project of Collaborative Innovation Center of Translational Medicine:
TM201827.

## Competing Interests

The authors declare that they have no competing interests.

## Author Contributions

- Jiaojiao Song conceived and designed the experiments, performed the experiments, analyzed the data, prepared figures and/or tables, authored or reviewed drafts of the paper, and approved the final draft.
- Junmei Zhou conceived and designed the experiments, authored or reviewed drafts of the paper, and approved the final draft.

## Animal Ethics

The following information was supplied relating to ethical approvals (i.e., approving body and any reference numbers):

All experimental procedures were performed in compliance with the guidelines on the ethical use of animals in Shanghai Jiao Tong University School of medicine DLAS (2019-B-033), as well as in conformance to international guidelines.

## Data Availability

The raw measurements are available in a Supplemental File.

## Supplemental Information

Supplemental information for this article can be found online at http://dx.doi.org/10.7717/peerj.8940#supplemental-information.

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
