# Peer review of "Effects of preservation duration at 4 °C on the quality of RNA in rabbit blood specimens"

_PeerJ, doi:10.7717/peerj.8940_

## Round 0.1 · original submission · Minor Revisions

Dear authors, please make corrections asked by Revewers 1 and 3, and answer objections of the Reviewer 2.

Reviewer 1 ·

Basic reporting

The manuscript entitled “Effects of preservation duration at 4 ℃ on the quality of RNA in rabbit blood specimens “ deals with the quality of RNA from whole blood and leukocytes isolated from rabbit blood in regard to prolonged storage. The obtained results suggest that the purity of whole blood RNA and leukocyte RNA does not indicate significant change after rabbit blood is stored at 4 ℃ for different preservation durations (from 1 hour to 7 days). The integrity of leukocyte RNA indicates the same result as above, but the integrity of whole blood RNA is significantly decreased after rabbit blood is stored at 4 ℃ for over 3 days. As for the expression of SMAD7, MKI67, FOS, TGFβ1 and HIF1α of whole blood RNA and leukocyte RNA it remained stable, but PCNA expression of whole blood RNA or leukocyte RNA is significantly decreased after rabbit blood is stored at 4 ℃ for over 24 hours or 7 days. Authors concluded that high-quality RNA is obtained from the fresher blood specimens, and if blood specimens are stored for over 3 days at 4 ℃, the quality of leukocyte RNA is more stable and of better quality than that of whole blood RNA.
This is a relevant methodological paper dealing with the storage conditions which present relevant and much needed data for this particular field.

Experimental design

Methods section is very well described with sufficient data for the study to reproducible. Since the study involves animal it seems that all the ethical standards were fully obeyed.
Please be consistent in the writing of either "hours" or "h" throughout the manuscript.

Validity of the findings

Data are very comprehensive and are graphically presents and easy to read.
Letters and numbers on the x and y axes are a bit small and difficult to read.

Additional comments

This is a very interesting methodological paper dealing with the impact of storage conditions of the quality of RNA.
This could be very interesting data for the ones working in the field of gene expression.
The first paragraph of the Discussion section is a bit of a repetition of what was already stated and could be shortened a bit.
If the Authors noted that that the mRNA of PCNA may be influenced by the preservation duration of blood specimens at 4 ℃ which obviously needs more research and they emphasize that it is something that should be done in the future, is it possible to do this additional experiments to resolve this issues already in this study and present this data in the submitted publication. It will certainly be an added value for the current manuscript.
Please use capital letter at the beginning of the first sentence of the Conclusion section.

Reviewer 2 ·

Basic reporting

The paper by Song and Zhou is an investigation on the feasibility of using blood samples preserved for different times at 4°C for biobanking and subsequent gene expression studies.
The language is correct, the introduction is clear and literature–based, structure is in line with Peer J guidelines and figures are clear and of good quality.

Experimental design

The present research fits with the scope of the journal.
The research question is well defined, however it is a rather well-addressed question.
The investigation was performed with good technical and ethical standards. Methods are described with a sufficient degree of detail.

Validity of the findings

Most of the papers on a similar topic use room temperature blood preservation after withdrawal (the so-called bench time), and this is generally less than 8 h (see for example Hebels et al, Environmental Health Perspectives volume 121, 4, 2013). Can the authors justify the use of 4°C? Furthermore, considering how valuable are samples to be biobanked, it is difficult to imagine that a researcher would accept the risk to maintain them for longer than 8 h without freezing them. It would definitely be more acceptable to freeze them within 8 h from withdrawal (possibly after fractionation). Indeed, the Authors show that RNA purity and integrity are not significantly modified up to 3 d for whole blood and to 7 d for leukocyte RNA; however, they recognize that RNA integrity is a parameter related to ribosomal RNA, and they show a degree of variability of specific mRNAs with time. Thus, it is possible that small but relevant differences in mRNA integrity take place during the time at 4°C, that could have an impact on future gene expression studies.
It is doubtful that the observed differences in the various tested mRNAs is due to gene expression changes taking place at 4°C, as the Authors seem to suggest in the discussion (lines 295-316). It appears more likely that some mRNA degradation can take place during this time, so that the expression “mRNA levels” should be used in place of “gene expression” here.

Additional comments

No further comment

Reviewer 3 ·

Basic reporting

Introduction: From line 74 the sentence should be excluded since it is not well formulated. It is necessary to explain what was the focus of the investigation and why that assessment was performed on rabbit blood instead of a human.



Discussion and conclusion: In lines 223 and 319, the sentences are not written appropriately. It should be reworded to be more understandable.
In line 324 the part of the sentence „but other studied genes were relatively stable“ should be replaced with „the expression of genes was stable“

Experimental design

Material and Methods:

Ethical approval is needed to be included in Material and Methods.
The authors have to explain why euthanasia methods are performed for the rabbits. Also, it is not clear if the same animal was used for successive blood sampling.

Validity of the findings

The paper describes diligent findings on information about effects of preservation duration at 4 °C for different preservation durations, from 1 hour to 7 days, on the quality of RNA in rabbit blood specimens. This paper addresses an interesting question: whether the quality of RNA from whole blood and/or leukocytes isolated from rabbit blood in regards to the effects of preservation duration at 4 °C could be a choice approach to valuable source of RNA for profiling gene expression.

Additional comments

Strong point:

The authors assessed that the purity of whole blood RNA and leukocyte RNA does not induce significant change after blood is stored at 4 °C for different preservation durations. Otherwise, the integrity of whole blood RNA and leukocyte RNA is degraded after a preservation duration for more than 3 days at 4 °C . The gene expression, especially of whole blood RNA, can decrease over the increased preservation duration.

---

## Round 0.2 · accepted · Accept

The paper can be accepted now. Congrats!